# Evolution of Urothelial Bladder Cancer in the Context of Molecular Classifications

**DOI:** 10.3390/ijms21165670

**Published:** 2020-08-07

**Authors:** Martina Minoli, Mirjam Kiener, George N. Thalmann, Marianna Kruithof-de Julio, Roland Seiler

**Affiliations:** 1Department of BioMedical Research, Urology Research Laboratory, University of Bern, 3008 Bern, Switzerland; martina.minoli@dbmr.unibe.ch (M.M.); mirjam.kiener@dbmr.unibe.ch (M.K.); George.Thalmann@insel.ch (G.N.T.); marianna.kruithofdejulio@dbmr.unibe.ch (M.K.-d.J.); 2Department of Urology, Inselspital, Bern University Hospital, 3008 Bern, Switzerland

**Keywords:** bladder cancer, muscle invasive, non-muscle invasive, molecular subtypes, evolution, targeted therapy, classification

## Abstract

Bladder cancer is a heterogeneous disease that is not depicted by current classification systems. It was originally classified into non-muscle invasive and muscle invasive. However, clinically and genetically variable tumors are summarized within both classes. A definition of three groups may better account for the divergence in prognosis and probably also choice of treatment. The first group represents mostly non-invasive tumors that reoccur but do not progress. Contrarily, the second group represent non-muscle invasive tumors that likely progress to the third group, the muscle invasive tumors. High throughput tumor profiling improved our understanding of the biology of bladder cancer. It allows the identification of molecular subtypes, at least three for non-muscle invasive bladder cancer (Class I, Class II and Class III) and six for muscle-invasive bladder cancer (luminal papillary, luminal non-specified, luminal unstable, stroma-rich, basal/squamous and neuroendocrine-like) with distinct clinical and molecular phenotypes. Molecular subtypes can be potentially used to predict the response to treatment (e.g., neoadjuvant chemotherapy and immune checkpoint inhibitors). Moreover, they may allow to characterize the evolution of bladder cancer through different pathways. However, to move towards precision medicine, the understanding of the biological meaning of these molecular subtypes and differences in the composition of cell subpopulations will be mandatory.

## 1. Introduction: The Clinical Problem of Bladder Cancer

Bladder cancer (BLCa) is one of the most common cancers of the genitourinary tract. It is the tenth most common cancer worldwide, accounting for 549,000 new cases and about 200,000 deaths per year [1]. The majority of BLCa are urothelial carcinoma, whereas squamous cell carcinoma, small cell carcinoma, adenocarcinoma and sarcoma are less frequent. In this review, BLCa refers to urothelial carcinoma and excludes the other rare forms. BLCa is around three times more common in men than in women and tobacco smoking is the main risk factor [2,3,4,5]. Family history increases the risk of BLCa and is mainly associated with early onset, however, only scarce data on genetically inherited BLCa is available [6,7,8]. BLCa is the costliest cancer to treat on a per-patient basis and causes large socio-economic burden [1,9,10,11,12]. Nevertheless, it is still relatively understudied [13].

BLCa has been classified into non-muscle invasive BLCa (NMIBC, 70–75% of patients) and muscle invasive BLCa (MIBC, 25–30% of patients), which are associated with distinct prognosis [14]. However, clinically and genetically very distinct tumors are summarized in NMIBC. Therefore, a definition of three different groups better accounts for the divergence in prognosis and treatment choice. The first group consists of NMIBC confined to the mucosa [14]. This group is divided into high- and low-grade pTa depending on cellular atypia and alterations in tumor architecture. Despite the distinct prognosis, the urothelial carcinoma in situ (CIS-stage pTis) is also part of the first group. CIS is defined as a high-grade, flat, non-invasive lesion confined to the mucosa with often vague borders and spreading across various areas of the bladder. The second group consists of NMIBC tumors that invade the submucosal layer, the lamina propria. These include stage pT1 and are of high-grade in most of the cases. Among NMIBCs, 70% are staged as pTa, 20% as pT1 and the remaining 10% as CIS lesions [15]. The third group consists of MIBC, either organ-confined, locally advanced and/or metastatic. MIBC tumors are staged from T2a to T4b depending on the extent of muscle invasion and are of high-grade in the majority of the cases.

The major clinical problem of NMIBC is the frequent short-term reoccurrence after the first resection (50–70%) [16,17]. Although most NMIBC have a low probability (10–20%) of progression and high survival rate (~90% 5-year overall survival) [16,17,18], lymph node metastases can be found in 5% of pT1 tumors, whereas 50% of CIS potentially progress locally and disseminate [19,20]. Around 10–20% of NMIBC-diagnosed patients progress to MIBC. Currently, these patients cannot be prospectively identified [17,21].

Contrarily, in MIBC, the rapid metastatic progression and the consequently high mortality of patients is of high clinical relevance [22]. The 5-year overall survival probability of 60% is dramatically reduced to less than 10% in case of early metastatic dissemination [23]. Currently, curative therapy options are missing and only a minority of patients with metastatic BLCa show long-term response to palliative treatments [24].

Physicians are often met with treatment failure and/or reoccurrence in all BLCa groups, indicating that the management of BLCa is complex and current classification systems do not depict the heterogeneity of this disease. Huge efforts are undertaken to understand the biology of BLCa. Tumor profiling on large-scale cohorts has dramatically improved our understanding of the biology of this disease and may allow us to identify molecular subtypes with a distinct clinical and molecular phenotype. The ultimate goal of these investigations should be to predict the likelihood of reoccurrence, response to treatment and progression early in patients’ history to guide clinical decision making.

In this review, we will present the state-of-the-art, clinical guidelines and standard operating procedures of BLCa and elucidate their limitations. We will also discuss how molecular subtypes discovered from the profiling of tumor samples improved our understanding of BLCa and their potential clinical meaning and application.

## 2. Current BLCa Management

### 2.1. NMIBC Treatment Options and Follow-Up

NMIBCs are first diagnosed with cystoscopy and confirmed by subsequent transurethral resection of the bladder tumor (TURBT) [25]. The clinical tumor, node and metastases (TNM) stages are determined based on cross-sectional imaging and pathological evaluation of the TURBT [25]. Furthermore, the histological grade is defined according to the 2004/2006 World Health Organization (WHO) classification [26].

Based on these evaluations, the European Organization for Research and Treatment of Cancer (EORTC) risk tables allow stratifying patients into low-, intermediate- and high-risk for reoccurrence or progression [17]. However, this may not be appropriate given that they are defined by different prognostic factors and require different follow-up strategies [27]. Prior disease reoccurrence rate and the number of distinct tumors are the strongest predictors of BLCa reoccurrence, whereas the most significant prognostic factors for disease progression and disease-specific survival are tumor stage and grade, and presence of CIS [17]. Ideally, a better differentiation, including prognostic markers, among these two risks (progression and reoccurrence) should be defined. Due to the high recurrence rate, patients undergo life-long monitoring. Standard of care includes cystoscopy every 3–6 months in the first 1–2 years and every year thereafter for 5 years, or lifelong [28,29]. In particular, intermediate- and high-risk tumors might reoccur even after 10 years, therefore a life-long surveillance schedule is recommended [30]. Treatment and surveillance of NMIBC patients is not only associated with considerable costs but also with high morbidity due to the considerable side effects, such as pain, risk for infection and irritation/damage of the urothelium.

Low-risk NMIBC patients after TURBT receive intravesical instillations of chemotherapy (Mitomycin C, Epirubicin or Pirarubicin) within 24 h after the surgery [25]. This targets the released tumor cells after TURBT, and significantly reduces the three-month reoccurrence risk [28,31]. Intermediate-risk and high-risk patients receive a Bacillus Calmette-Guérin (BCG) vaccination, which is an attenuated strain of *Mycobacterium bovis*. This prolongs reoccurrence time decreasing disease progression [28,32,33,34]. BCG is not always well tolerated, and the risk of relapse varies between 30% and 40% [35].

### 2.2. MIBC Treatment Options and Follow-Up

Neoadjuvant cisplatin-based chemotherapy (NAC) prior to radical cystectomy is the golden-standard of care for all non-metastatic MIBC patients [36]. This treatment should be offered to all patients with non-metastatic BLCa, organ-confined (cT2N0M0) or locally advanced BLCa (cT3a-T4a, N0-NX, M0) [23]. The systemic treatment for NAC can be the combination of methotrexate, vinblastine, doxorubicin and cisplatin (MVAC) or the combination of cisplatin with gemcitabine (GC), which is used because of the lower toxicity but comparable efficacy [36,37]. Life expectancy is increased in 40% of NAC-treated patients [14,29,38], which can be attributed to the reduction of tumor size prior to surgery, the treatment of micrometastases and the prevention of metastasis formation [36]. While only 10% of MIBC patients present metastasis at diagnosis, 40–50% show metastatic formations after reoccurrence detected during follow-up after cystectomy. For these patients, an effective therapy is lacking and the treatment intent is palliative. First-line treatment is cisplatin-based chemotherapy [14,29,36], and second-line treatments for cisplatin-resistant and metastatic BLCa are the Food and Drug Administration(FDA)-approved immune checkpoint inhibitors (ICIs) which target PD-L1 (atezolizumab, avelumab and durvalumab) or PD-1 (nivolumab and pembrolizumab). However, only 20–25% of patients respond positively to treatment [14,39,40,41].

Novel targeted therapies have recently been approved for treatment of patients resistant to cisplatin-based chemotherapy and ICIs. Erdafitinib, an inhibitor of the tyrosine-kinase activity of fibroblast growth factor receptor (FGFR), is approved for patients with FGFR3 or FGFR2 alterations [42,43]. Finally, enfortumab vedotin, an antibody-drug conjugate that targets Nectin-4, has shown promising response rates [44].

Follow-up includes, in the three years after initial treatment, computed tomography (CT) scan of chest, abdomen and pelvis every 3–6 months, and reoccurrence is treated with radiotherapy.

Importantly, the survival rate of MIBC patients has remained unchanged for the last 30 years [5] and currently, the treatment is offered in “a one size fits all” manner and by trial and error. The response is documented during the follow-up and only a minority of patients show a positive treatment response (e.g., NAC 30–40%, ICI 20–25%). Therefore, selecting patients according to the likelihood of response might help to improve patients’ outcomes and avoid overtreatment in likely non-responders. The identification of accurate and predictive biomarkers for non-responders vs. responders is one of the major challenges in BLCa research.

## 3. Lack of Predictive Biomarkers: First Cause of Treatment Failure

The main problem for NMIBC is the high recurrence rate and early identification of tumors with the potential to progress.

The risk of reoccurrence after installation of chemotherapy varies between low-, high- and intermediate-risk tumors but also between immediate and delayed treatments. For low-risk tumors, the 5-year reoccurrence risk does not change between immediate (43%) or delayed treatment (46%) [45]. In contrast, immediate treatment decreases the recurrence risk from 31% to 20% for the intermediate-risk tumors and from 35% to 28% for high-risk tumors [45]. Therefore, rapid selection of patients according to the likelihood of response is crucial to improve chemotherapy efficacy.

Although BCG therapy has been used for more than 30 years, its mechanism of action is still under investigation. About 40% of patients fail BCG therapy and tumor reoccurs within 2 years [14,46]. BCG failure refers to three different conditions: (1) BCG relapse, if the disease reoccurs after ≥6 months of disease-free survival, (2) BCG refractory, if the tumor persists after 6 months of treatment and (3) BCG intolerance, if the treatment has been discontinued due to toxic side effects of BCG [14,28,47]. Many factors can cause BCG failure, e.g., the presence of metastasis prior to therapy, inappropriate immune response or an adaptive immune resistance with overexpression of exhaustion markers (PD-1 and PD-L1) [48,49,50]. Although there is no standard of care for patients after BCG failure, radical cystectomy is considered the preferred treatment [25]. Alternative approaches, such as immune checkpoint inhibitors against PD-L1 or PD-1, have been taken into consideration as therapy in combination with BCG, given the role of exhaustion markers (PD-L1 and PD-1) in BCG resistance and relapse [51]. However, efforts to understand how to identify patients that will benefit from BCG, combination therapies or will have to undergo radical cystectomy are much needed [52].

Such predictive biomarkers are also missing for MIBC patients. The major clinical problem in MIBC is treatment failure due to inaccurate treatment selection, which is associated with the high mortality of MIBC patients. This is exemplified by a complete NAC response only in a minor fraction of patients. Several clinical trials have shown that NAC improves overall survival in only 30–40% of patients [38,53,54,55,56,57]. In NAC non-responder patients, the treatment delays the effective therapy, directly associated with adverse prognosis, and overtreats patients that will suffer from unnecessary side effects [58,59,60].

The mechanisms of resistance of MIBC to systemic treatment are largely unknown and molecular biomarkers for the selection of effective second-line treatments are currently lacking. A one treatment fits all approach for such a highly heterogeneous disease is certainly a limiting factor. Stratifying patients into “subtypes” based on the molecular landscape can be used to predict the response to treatment [61,62,63]. BLCa can be grouped into at least two molecular subtypes: basal/squamous-like and luminal subtypes, and some of these subtypes were associated with therapy response. McConkey and colleagues employed gene expression profiles of NAC-treated MIBC tumors to predict therapy response. They showed that basal tumors were associated with better survival in the context of NAC and that tumors expressing wild-type (WT) p53-associated gene expression (p53-like subtype) were associated with bone metastasis and chemo-resistance [61]. In a retrospective study, Seiler and colleagues investigated the association between molecular subtypes and the response to NAC. They proposed four molecular subtypes (Basal, Claudin-low, luminal and luminal-infiltrated subtypes) that can predict NAC response and confirmed that basal tumors were more chemo-sensitive and benefited the most from NAC compared to cystectomy only. In addition, they showed that irrespective of the treatment, the luminal subtype had better overall survival. On the contrary, the claudin-low subtype, defined as a subset of basal tumors with (1) a decreased expression of claudin-3 and claudin-4, (2) an enrichment for tumor-initiating cells (TIC) signatures and (3) an over-expression of epithelial-mesenchymal transition (EMT) transcription factors (e.g., SNAI1, TWIST1), had the worst overall survival and did not profit from NAC [62]. In the context of immunotherapy, ICIs are effective only in a subset of patients (20–25%) [39,40,41]. Mariathasan and colleagues performed transcriptomic analysis of a cohort of MIBC tumors treated with atezolizumab, an anti-PD-L1 drug. The responders were correlated with high neoantigen or tumor mutation burden, and with infiltration of CD8^+^ T-effector cells. Non-responders were correlated with a gene expression signature of transforming growth factor β (TGF-β) signaling in fibroblast, especially in cases where CD8^+^ T-effector cells were only present in the peritumoral stroma and not in the tumor parenchyma [63]. However, clinical trials testing these stratifications of patients into “subtypes” are needed.

In conclusion, treatment failure is frequent in both NMIBC and MIBC. This may indicate that the current classification systems do not depict the heterogeneity of BLCa and that treatment itself does not consider BLCa heterogeneity. Moreover, current pathological assessment is inaccurate and around 40% of tumors are clinically up-staged, which leads to an inadequate treatment choice [64,65,66]. Improvement of patient stratification to guide treatment options may be achieved using molecular classifications, and one way to further classify BLCa patients is according to molecular subtypes. Understanding of their biological role and therapy response is mandatory for clinical implementation.

## 4. Molecular Classification

### 4.1. NMIBC and MIBC: Two-Pathway Theory

Histological and cytogenetic analysis of BLCa tumors revealed that low-grade NMIBC tumors had few cytogenetics changes, while MIBC tumors were more genetically unstable [67]. Therefore, based on the fact that NMIBC and MIBC tumors present with different clinical behavior, histology and evolution, two distinct carcinogenic pathways, the papillary and the non-papillary pathway, have been proposed. The papillary pathway includes hyperplasia which will give rise to low-grade NMIBC; in the non-papillary pathway, instead, flat dysplasia and/or CIS are believed to be the precursors of MIBC tumors. However, high-grade NMIBCs seem to derive from the co-occurrence of hyperplasia and dysplasia, thus suggesting that the two-pathways intersect.

The papillary pathway is characterized by activation or overexpression of oncogenes leading to the genomic stable low-grade non-invasive papillary (Ta) tumor. FGFR3-activating mutations are present in almost all the Ta tumors but absent in CIS and less frequent in high-grade T1 NMIBC (30%) and MIBC (10%) [68,69,70,71,72]. Therefore, they are considered as driving alterations that induce the hyperplasia and tumorigenesis of the papillary pathway [68,70,71,72,73,74]. Activating FGFR3 mutations are associated with good prognosis and low recurrence rate. Mutations in the telomerase reverse transcriptase (TERT) gene promoter, an early genomic alteration associated with predisposition to develop BLCa, are present in the majority of tumors carrying FGFR3 mutations [75]. TERT re-activation may prevent senescence in FGFR3 mutant hyperproliferative tumors and protect genomic integrity. In normal conditions, FGFR3 regulates the activation of RTK/RAS/RAF/MAPK and PI3K/Akt/mTOR pathways. Therefore, in BLCa activation RTK/RAS/RAF/MAPK pathway was proposed as a likely mechanism that regulates cell growth in low-grade NMIBC [74,76]. Both activation of FGFR3 and RAS results in stimulation of the RTK/RAS/RAF/MAPK pathway, supporting the reason why in 10–15% of NMIBC, FGFR3 and HRAS/KRAS mutations are mutually exclusive [77,78]. Conversely, concerning the PI3K/Akt/mTOR pathway, activating mutations in PI3KCA are enriched in FGFR3 mutant tumors and present in 25% NMIBC and in 20% MIBC [68,79,80,81]. However, PIK3CA mutations are associated with low risk of progression [82]. Alterations in chromatin-modifying genes are frequent in all stages of BLCa, indicating that they can be early genomic alterations [68,70]. The lysine demethylase 6A (KDM6A) is mutated in 60% of low-grade Ta tumors and inhibits H3K27 methyltransferase EZH2 [70,79,83]. Given that EZH2 is often overexpressed and associated with MIBC, KDM6A activity might counteract tumor progression [84]. Moreover, truncating mutations in the STAG2 gene, which encodes a component of the cohesin complex, are highly frequent in low-grade Ta tumors and associated with low risk of progression and recurrence [82,85].

The most common copy number alteration (CNA) in Ta tumors is loss of heterozygosity (LOH) of 9q and 9p [79,82,86,87,88]. LOH 9 leads to deletions of several tumor suppressors genes, such as CDKN2A (9q), TSC1 (9p) and PTCH1 (9p) [86,87,89]. Loss of CDKN2A is present in more than 90% of FGFR3-mutant MIBC and homozygous deletions of CDKN2A are associated with a higher grade and higher risk of progression [70,82,90,91]. Moreover, CDKN2A gene encodes p16^INK4a^ and p14^ARF^, which induce cell cycle arrest via TP53 and RB1 signaling pathways [92]. In BLCa, CDKN2A deletion promotes proliferation and is mutually exclusive with TP53 and RB1 loss [68,92,93].

The genomic and transcriptomic landscape of low-grade Ta NMIBC indicates an important role for sustained proliferative signaling and is characterized by more clearly distinct mutations than MIBC. However, high-grade NMIBC shares certain characteristic genomic alterations with MIBC.

The non-papillary pathway gives rise to the majority of MIBCs. They share some characteristic mutations, CNA and chromosomal abnormalities resulting from defects in DNA replication/repair machinery genes and mutations in tumor suppressors genes [21]. Tumors with high genomic instability are associated with advanced stage and poor survival [88]. CIS might give rise to MIBC through the non-papillary pathway by the gradual accumulation of genomic abnormalities [88,94].

Mutations in tumor suppressor genes such as TP53, RB1 and PTEN are frequent in MIBC [21,95,96]. TP53 mutations are thought to have a central role in the non-papillary pathway and they are present in 50% of MIBC and 10–20% of high-grade T1 tumors [68,69,70,95,97]. MDM2, which negatively regulates p53, is overexpressed in 30% of MIBC and it is mutually exclusive with TP53 mutations in the majority of tumors [68]. In some tumors, the loss of function of p53 co-occurs with the loss of function of Rb1 [92,96,98,99,100]. The level of Rb1 is known to be inversely correlated to p16 (CDKN2A) level, and alterations in the Rb1/p16 tumor suppressor checkpoint pathway are associated with MIBC and increased risk of progression [95]. In some cases, Rb1 function is inhibited due to the overexpression of cyclin D1 (CCND1) or loss of CDKN2A, which are prevalent in high-grade tumors [68,70,97,101]. Therefore, despite loss of the RB1 gene in only 10% of MIBC, the Rb1 pathway is potentially dysregulated in a much higher fraction of tumors. The heterogeneous mechanisms resulting in a similar phenotype are part of the problem with treatment failure, resistance and lack of biomarkers.

Defects in DNA damage response genes are present in around 30% of MIBC and high-grade NMIBC, whereas they are absent in most of low-grade NMIBCs [102]. The most frequently altered gene is ERCC2, a component of the nucleotide excision repair machinery (NER). It is mutated in high-grade T1 tumors (17%) and MIBC (20%), but wild type (WT) in Ta [70,79]. Interestingly, mutations in NER pathway genes can occur in different types of cancer, but recurrent mutations are rare—ERCC2 in BLCa is an exception [103,104,105]. The Cancer Genome Atlas (TCGA) consortium was identified and described an ERCC2 signature and confirmed its association with smoking [68,92]. The function of ERCC2 in the urothelium is not known but given its association with smoking, it can be hypothesized that it protects against damage induced by carcinogenic metabolites accumulating in the urine.

Besides the induction of genetic instability, several genetic alterations associated with MIBC regulate cell growth and invasion. EGFR family genes such as EGFR and HER2 (ERBB2) are frequently overexpressed in CIS, MIBC and especially in metastatic BLCa [68,70,106]. Activation of EGFRs induce activation of RAS/MAPK and PI3K/AKT/mTOR pathways. Moreover, as transcription factors, EGFR family members can induce the expression of proliferation promoting oncogenes such as MYC and CCND1 or genes regulating migration/invasion such as COX2 and MMPs [107,108,109]. HER2 amplification is expressed higher in lymph node metastases than in primary tumors and both HER2 and EGFR overexpression are associated with a higher risk of recurrence and progression to invasive disease [69,110,111,112,113]. In contrast to HER2 and EGFR, ERBB3 overexpression has been linked to low-grade papillary NMIBC and good prognosis [69,110,111,112].

Alterations of the PI3K/AKT/mTOR pathway are involved in the progression of MIBC to metastatic disease [114,115]. PTEN, a negative regulator of PI3K, is downregulated in over 90% of MIBC and in 40% of NMIBC. MIBC frequently harbor mutations in PTEN, whereas Ta and T1 tumors retain WT PTEN [81,114,116]. Loss of PTEN is associated with aggressive MIBC and metastasis. In more than 40% of MIBC, loss of PTEN co-occurred with TP53 loss [117]. Loss of PTEN and TP53 is associated with poor survival [81,117], suggesting that the dual loss cooperates in tumor invasion and metastasis formation [117].

The genomic and transcriptomic characterization of BLCa revealed distinct molecular alterations in low-grade non-invasive papillary tumors and MIBC. However, these studies additionally discovered similarities between high-grade NMIBC and MIBC. As MIBCs, high-grade NMIBCs have more complex CNAs and they gradually accumulate genomic instability during progression compared to low-grade Ta tumors [71,79]. Mutations affecting DNA replication/repair machinery genes and tumor suppressors genes are shared among high-grade papillary tumors (Ta/T1) and MIBC developing via the non-papillary pathway. These alterations lead to the acquisition of the invasion potential in high-grade papillary tumors and can be the intersection between the two distinct carcinogenic pathways [69]. This indicates that the two pathways of BLCa development can intersect. Accordingly, 10–20% of MIBCs originate from the progression of high-grade NMIBCs.

### 4.2. Molecular Subtypes to Understand BLCa Biology

The two-pathway theory can be used to understand early diseases but, given the complexity and high heterogeneity of BLCa, it cannot explain the biology and progression of some tumors. Tumor profiling on large-scale cohorts, instead, allows us to identify molecular subtypes of BLCa with distinct clinical and molecular phenotype (Figure 1). This approach can improve our understanding of the biology of this complex disease. Multiple studies have classified BLCa into molecular subtypes based on transcriptomic data. The different subtypes and their clinical relevance are discussed in the following.

Hedegaard et al. identified three molecular subtypes of NMIBC (Class I, Class II and Class III) associated with distinct basal and luminal features and disease outcome (Table 1) [69]. Class I, with luminal-like features, associated with a good prognosis, is characterized by elevated expressions of early cell cycle regulators (CCDN1) and FGFR3 mutations. Given its features, Class I is comparable to the Urobasal A subtype (UroA) proposed by Sjödahl and colleagues who analyzed a mixed cohort of NMIBC and MIBC tumors [98]. Uro A was represented in the majority of the cases in low-grade non-invasive tumors. Class II, with luminal-like features, is characterized by elevated expressions of late cell cycle regulators and transcription factors associated with epithelial-to-mesenchymal transition (EMT) and ERBB2. Class II is comparable to the genomic unstable (GU) subtype from Sjödahl and colleagues. Class III, with basal-like features, associated with shorter overall survival, represents a dormant tumor state of NMIBC which shifted towards a basal phenotype with FGFR3 mutations. Only Class II and III tumors are associated with high stage and grade, high risk (according to EORTC), presence of CIS and progression to MIBC. Therefore, non-invasive tumors of Class II are associated with a high risk of progression through the non-papillary pathway (Figure 1) [69]. In contrast, class I tumors become muscle invasive via the papillary pathway, but this rarely happens. Interesting, it seems that Class III tumors switched to Class II upon progression. Therefore, given the fact that Class III tumors have a basal phenotype and FGFR3 mutations, it is not correct to associate the papillary pathway exclusively to luminal-like tumors. However, the origin of Class III tumors needs to be investigated.

It has been proven that NMIBC progression is also influenced by tumor differentiation status [118]. Therefore, NMIBC subtypes can represent three different development pathways of NMIBC that better differentiate between tumors with high and low risk of progression potential.

Currently, there is limited data on NMIBC molecular subtypes. Implementation of subclass stratification in NMIBC patients could facilitate biomarker discovery and ultimately assist in the identification of patients at higher risk of recurrent and/or progressive disease.

Concerning MIBC, to date, several groups have proposed comparable molecular classifications that have been associated to specific biological features, clinical outcomes and also responsiveness to therapy (Table 2) [62,68,91,92,98,118,119,120,121,122,123]. A consensus classification has been introduced to facilitate the implementation of molecular subtypes of MIBC in clinical trials and clinical practice [96]. The consensus classification is derived from the combination of the analysis of six previously published classifications of MIBC (Baylor [118], University of North Carolina (UNC) [121], Cartes d’Identité des Tumeurs (CIT)-Curie [91], MD Anderson Cancer Center (MDA) [120], Lund [123], TCGA [92]) with public transcriptome data of MIBC. This resulted in the description of six molecular subtypes for MIBC: luminal papillary (LumP), luminal non-specified (LumNS), luminal unstable (LumU), stroma-rich, basal/squamous (Ba/Sq) and neuroendocrine-like (NE-like) [96].

The Ba/Sq subtype is the most prevalent (35%) [96], characterized by a basal and squamous differentiation. Ba/Sq tumors express basal and stem cell markers such as KRT5/6, KRT14 and CD44, normally expressed in the basal or stem cell layer of the urothelium containing stem cell subpopulations [98,119]. It is strongly associated with STAT3 regulon activation, elevated HIF1A and downregulation of genes associated with urothelial differentiation such as FOXA1, GATA3 and PPARG [96]. Frequently mutated genes include the cell cycle regulator TP53 (61%), generally altered in advanced MIBC and with a central role in the non-papillary pathway [95,96]. TP53 is associated with a high risk of T2+ stage [124] and Ba/Sq subtype is over-represented in higher clinical stages (T3/T4) [96]. In 25% of Ba/Sq tumors, RB1 was mutated and TP53 and RB1 mutations co-occurred in 14% of Ba/Sq tumors, suggesting an interaction between the two cell cycle regulators. However, it was shown in mouse models that the dual loss of TP53 and RB1 is necessary but not sufficient to induce the invasion pathway in BLCa [125]. Therefore, given the prevalence of TP53 and RB1 mutations, Ba/Sq tumors originate from the non-papillary pathway, however other mutations sustain the invasion potential (Figure 1).

Concerning invasion and migration of bladder cancer cells, the interaction between cancer cells and the microenvironment is important. EMT is known to be an important process during tumor cell invasion. Downregulation of the cell adhesion molecule CDH1 (E-cadherin), and upregulation of its transcriptional repressors, such as ZEB1, ZEB2, SNAIL-1 and TWIST, in BLCa is associated with a poor prognosis and high risk of metastasis in BLCa [120,126]. Interestingly, patients with basal tumors at diagnosis present a more invasive and metastatic disease with a high EGFR activity and upregulation of EMT markers, such as ZEB1, ZEB2 and Vimentin [68,92,120,126]. It is known that EGFR activity can induce invasion by activating STAT3 which induces Twist gene expression [91,127]. The Ba/Sq subtype comprises a subset of Claudin-low tumors that is not included in the consensus classification but has been described in previous studies [62,122]. The Claudin-low subtype over-expresses transcription factors promoting EMT, resulting in a pronounced mesenchymal signature, and it is associated with poor survival irrespective of treatment [62,122]. Claudin-low tumors fundamentally differ from Ba/Sq tumors in their response to NAC, highlighting the importance of molecular subsets within subtypes of the consensus classification. The Claudin-low subtype is comparable to the Ba/Sq-infiltrate subtype proposed by Marzouka and colleagues which expresses CDH3 (P-cadherin) and EGFR but no ERBB2 or ERBB3 [123]. Similarly, Claudin-low tumors exhibit features of the Ba/Sq and stroma-rich subtype of the consensus classification. Both subtypes show higher immune infiltration compared to other subtypes [96]. In addition, the stroma-rich subtype exhibits an extraordinarily high contribution of stroma cells to the tumor mass. However, it is not yet fully understood whether the stroma-rich subtype classifies separately from Ba/Sq tumors due to the contamination with stroma cells or is biologically distinct from the Ba/Sq subtype. The stroma-rich subtype is better described later in the review. Given the differences in prognosis and response to therapy, functional comparisons between Claudin-low tumors and Ba/Sq tumors would help us to understand the most important pathways involved in the diversion of these two basal subtypes. This would facilitate the discovery of targeted treatments to block the rapid progression to metastasis formation of a subset of tumors sharing a basal phenotype.

Researchers aimed to identify transcription factors involved in basal expression signatures that determine the basal-like phenotype. They proposed ΔNp63 to be involved in the regulation of basal transcription programs. ΔNp63α, an isoform of p63 belonging to the p53 family, regulates epithelial development and stem cell biology and has also been implicated in basal breast cancer [120]. In BLCa, it was shown that ΔNp63α induces the expression of miR205, which inhibits EMT [128]. Downregulation of p63, as observed in MIBC, might therefore contribute to invasion [128]. However, high levels of ΔNp63α were associated with a lethal group of MIBC [129,130]. Both, STAT3 and EGFR activity seem to promote ΔNp63α expression [131]. The role of ΔNp63α in MIBC is not clear but it is probably involved in cell proliferation rather than in invasion. Moreover, p63 is highly expressed in the epithelial subset of BLCa and may therefore be involved in the biology of MIBC with epithelial and stemness phenotype [128]. Further functional investigations of the two major p63 isoforms (TA and ΔN) are needed to understand the p63–EMT relationship and to unravel the prognostic and therapeutic value of p63 in basal subtypes.

Luminal-like tumors (LumP, LumU and LumNS) presented a papillary morphology enriched in luminal markers such as low molecular weight KRT20 and uroplakins such as UPK1A and UPK2, normally expressed in terminally urothelial cells. Luminal MIBC exhibit gene expression signatures of PPARG, GATA3 and FOXA1 transcription factors involved in urothelial differentiation, and regulatory factors and receptors of the estrogen signaling (ESR2) [96,120]. Discrepancy in outcome of patients with luminal-like tumors has been reported [92,121,132]. However, mutations status and oncogenic mechanism were different between the three luminal subtypes, indicating that non-invasive luminal tumors can progress to invasion through different carcinogenic pathways and not only through the papillary pathway. The investigation of cellular mechanisms contributing to the differences between luminal subtypes would elucidate these distinct pathways of progression.

The LumP subtype was the most prevalent (24%), with the best overall survival compared to the other luminal subtypes. LumP tumors were strongly associated with high FGFR3 activity caused by gene fusion, mutation or amplification [96]. FGFR3-activating mutations and overexpression are predominantly associated with low-grade Ta tumors developing via the papillary pathway, which have a favorable outcome in most cases [95,133,134,135]. The LumP subtype is comparable to the Urothelial-like subtype of Sjödahl and colleagues formerly named “Urobasal” (Uro A and Uro B) [71,123]. Uro A is comparable to Class I NMIBC (luminal-like) [69]. The Uro B subtype was predominant in MIBC tumors and they are believed to be the progressed version of Uro A tumors. Comparable to UroB tumors, LumP tumors are believed to derive from Ta/T1 tumors. Given the elevated expression of FGFR3 and the homozygous CDKN2A deletions in 33% of LumP tumors, we can assume that LumP tumors represent the fraction of luminal FGFR3-mutant tumors that originated from Class I tumors and progressed through the papillary pathway (Figure 1) [121].

The LumP tumors are thought to have a low likelihood of response to cisplatin-based NAC in contrast to basal tumors [62]. Interestingly, Choi and colleagues described the existence of a portion of luminal tumors (p53-like subtype) which were associated with WT TP53 gene expression signature and were resistant to chemotherapy. However, it is known that p53 level increases in response to DNA damage to stop cell cycle and induce apoptosis [120]. Alternatively, WT p53 induced reversible senescence which impaired the apoptotic response after chemotherapy-induced DNA damage, as it was shown in mouse models for breast cancer [136]. The molecular basis of a positive or negative p53-signature in the context of NAC is not known in BLCa. In the analysis conducted by The Cancer Genome Atlas (TCGA) Consortium, it was shown that MDM2 amplification or overexpression was predicted to inactivate p53 in 76% of MIBC [68]. Investigation of mechanisms of chemotherapy resistance in this portion of luminal tumors is needed.

The LumU subtype instead was associated with worse overall survival compared to the other luminal subtypes. It is the most genomic unstable subtype among luminal subtypes and harbors the highest load of apoliprotein B mRNA-editing enzyme catalytic polypeptide-like (APOBEC)-induced mutations [96]. LumU had higher cell cycle activity than other subtypes and it was enriched in ERCC2 and TP53 mutations. The majority of NMIBC (85%) with mutated p53 are of high grade, indicating that p53 inactivation occurred in the majority of NMIBC that have the potential to invade [137]. The LumU subtype was associated with an overexpression of HER2, frequently observed in CIS, MIBC and metastatic BLCa [68,70,106]. The LumU subtype corresponds to the genomic unstable subtype, discovered by Sjödahl and colleagues. According to the fact that LumU tumors are enriched in mutations characteristic of the non-papillary pathway, we can assume that LumU/GU tumors originated from Class II NMIBCs through the non-papillary pathway (Figure 1) [69]. Given that LumU and LumP tumors progressed through distinct pathways, as shown by their distinct genomic properties, it is important to separate these tumors and characterize them distinctly.

The last luminal subtype proposed in the consensus classification was described as a non-specified luminal (LumNS) subtype and includes a minor fraction of tumors (8%). The LumNS subtype is characterized by an elevated stromal and moderate immune infiltration. It is enriched in mutations effecting regulatory factors such as ELF3 and PPARG. The LumNS is associated with the worst overall survival in the luminal group. It is comparable to the epithelial-infiltrated subtype of Sjödahl and colleagues, which combines features of the GU and Uro subtypes [98]. Accordingly, LumNS exhibits features of the LumP and LumU subtypes in the consensus classification.

An additional subtype, the stroma-rich subtype, is also highly infiltrated by stromal and immune cells. The stroma-rich subtype was associated with gene expression signatures of smooth muscle, endothelial cells, fibroblasts and myofibroblasts. Both LumNS and the stroma-rich subtypes were not well defined and may probably be the result of cluster analysis. Advanced MIBC tumors are composed of tumor cells and stromal and immune cells. This can confound global gene expression analysis and bias the results towards stromal gene signatures. To resolve this problem, Sjödahl and colleagues histologically analyzed MIBC tissue in addition to mRNA sequencing approaches [98,123]. They showed that the type and the level of infiltrating non-tumor cells can vary between tumors sharing the same tumor cell phenotype [98,123]. Therefore, LumNS and the stroma-rich subtypes represent a heterogeneous class of tumors that we need to investigate at a higher resolution and ideally, by functional assays, to understand their role and origin.

The neuroendocrine-like (NE-like) subtype is the rarest of the consensus classification and only found in 3% of MIBC. NE-like tumors are enriched with neuroendocrine differentiation markers including synaptophysin, chromogranin A (CGA) and neuron-specific enolase (NSE or CD56). Moreover, they are characterized by high cell cycle activity and the co-occurrence of TP53 and RB1 mutations or deletions [92,96,98]. Patients with tumors carrying alterations in both TP53 and RB1 have a high risk of BLCa progression and worse clinical outcome than patients with only one of these gene alterations [95]. The majority of NE-like tumors display partial or complete neuroendocrine histology. Given the large contribution of neuroendocrine histological variant in the NE-like subtype, it is not surprising that this subtype is associated with the worst prognosis. Neuroendocrine tumors comprise small cell carcinoma of the bladder (SCCB) and large cell carcinoma of the bladder (LCCB). SCCB is more prevalent than LCCB but is extremely rare, making up only 0.5% of BLCa, and it is usually diagnosed at late stages [138]. To date, no guidelines for the treatment of neuroendocrine tumors have been formulated. Treatment strategies commonly follow the therapy approaches for small cell carcinoma (SCC) of the lung, including surgery, chemotherapy and radiation [139]. However, this is clearly not sufficient to treat this most aggressive form of BLCa with a 5-year overall survival rate of only ~20% [138]. Very little is known about the biology and evolution of neuroendocrine tumors. However, recent evidence supports the idea that SCCB originates from urothelial cells. How urothelial cells transdifferentiate remains elusive [138,140]. Possibly, anti-cancer treatments can induce the transdifferentiation of urothelial cells into tumors with a neuroendocrine phenotype in a similar manner as it has been described for neuroendocrine transdifferentiation of prostate adenocarcinoma upon androgen-deprivation therapy [141]. A better understanding of the pathogenesis and molecular differences between the NE-like phenotype/SCCB and the other BLCa subtypes may serve to understand their different biological meaning and origin. However, NE-like tumors are rare, impeding large-scale genomic and transcriptomic analysis of patient datasets. In order to characterize this tumor subtype, it is mandatory to create tumor models reflecting the molecular features of NE-like tumors.

## 5. Molecular Classification to Guide Treatment Choice

Molecular subtypes could be used to stratify patients and may help to identify patients whom would benefit from conventional, targeted or combined therapies (Figure 2) [91,120].

Regarding conventional therapies, basal tumors, should be selected for NAC and not selected for radiotherapy given their hypoxic microenvironment (Figure 2) [62,126,142,143]. Radiotherapy, given the lack of predictive biomarkers, is underutilized and mainly used as palliative therapy. Recent findings suggest that tumors with an increased expression of genes associated with T-cell activation and the INF*γ* signaling pathway would benefit from radiotherapy [144]. However, clinical trials to analyze the impact of subtyping on response to radiotherapy are needed.

In the context of immunotherapy, no subtypes proposed in the consensus classification have a profile (infiltration of CD8^+^ T cells, high IFN*γ* signaling and low TGFβ pathway) associated with a positive response to ICIs [63]. However, high infiltration of CD8^+^ T cells has been associated with Claudin-low tumors [62,122] and luminal-infiltrate tumors [62]. Therefore, the authors hypothesize that these patients can benefit from checkpoint inhibitors (Figure 2) [39,40,145]. Interestingly, it was shown that NE-like tumors, given the low TGFβ pathway (low TGFβ1 and TGFβR1), and LumU, given the highest tumor mutation burden and tumor antigen burden, would benefit from ICIs (Figure 2) [146]. Stroma-rich tumors, instead, given the high activity of the TGFβ pathway, would probably be resistant to ICS [146]. More clinical trials to analyze the impact of subtyping on response to ICIs are needed.

There is evidence that a fraction of BCLa patients might respond to targeted therapy [97]. EGFR is frequently overexpressed in Ba/Sq tumors and exerts an oncogenic function. Therefore, selecting these tumors for EGFR-targeted therapies seems to be promising (Figure 2). Monoclonal antibodies or tyrosine kinase inhibitors (TKI) targeting EGFR are effective in different cancers like lung, head, neck and colorectal [147]. In contrast, targeting EGFR in BLCa showed conflicting results [148,149,150,151,152,153]. However, patient selection was not conducted according to EGFR status in some of these clinical trials, which may have concealed the response to EGFR-overexpressing tumors. Currently, afatinib, a second-generation of tyrosine-kinase inhibitor of ERBB2 and EGFR, is being tested in a clinical trial phase II for metastatic, chemotherapy refractory BLCa patients with HER2/ERBB3 mutations and HER2/EGFR amplification [154]. Notably, resistance to TKIs can occur by secondary mutations, activation of alternative pathways or anomalies in the downstream signaling transducers [155]. In the case of EGFR, it was shown in colon cancer that RAS mutations (KRAS and HRAS) can inhibit the response to EGFR-targeted therapy [91,156]. Therefore, the interaction between RAS mutation status and EGFR-targeted therapy should be further investigated. Combination targeted therapy should be considered in case of a Ras-mediated resistance to EGFR inhibitors [91].

In contrast to other solid cancers, targeting HER2 in MIBC did not show promising results and no HER-2 targeting drug has been approved yet [148,157]. Most likely, this is due to the fact that the majority of clinical trials evaluating HER-2 inhibitors in BLCa did not select patients based on HER-2 status. Those trials that did evaluate HER2 status mostly used immunohistochemistry (IHC) and/or fluorescence in situ hybridization (FISH). However, it was proven that patient selection based on DNA and RNA sequencing techniques better stratifies responders and non-responders [158,159]. LumU tumors overexpressing HER2 can be used to discover biomarkers to select patients that would probably benefit from HER-2-target therapy.

FGFR3 overexpression is believed to be the oncogenic mechanism in LumP tumors and can be considered a potential therapeutic target for these tumors (Figure 2) [96]. Erdafitinib, an inhibitor of the tyrosine kinase activity of FGFR, was recently approved as a treatment for patients with chemo-resistant locally advanced or metastatic BLCa carrying FGFR3 or FGFR2 alteration [42,43]. Given that LumP tumors would probably have a low likelihood of response to cisplatin-based NAC [62], they might be selected for FGFR3-target therapy.

## 6. Limitations and Perspectives of Molecular Classification

Current findings from the molecular investigation of BLCa cohorts showed that patients can be grouped into classes based on their molecular landscape. Of utmost importance, these molecular classes have been established based on investigations of mainly untreated tumor samples. However, the molecular classification has some limitations given the fact that BLCa is a very heterogeneous disease exhibiting vast histological and molecular intra- and inter-tumor heterogeneity was shown in some tumors [160,161]. Consequently, the existence of multiple subtypes within the same tumor has been demonstrated. Therefore, one molecular subtype cannot be representative for these heterogeneous tumors. Further investigations to study the impact of multiple subtypes on tumor heterogeneity and response to treatments are needed. There is current standardization and consensus on gene expression-based molecular subtyping [96]. However, further standardizations are needed for the implementation in the clinical setting. In addition, other analytical aspects such as tumor cell content, immune cell infiltration, selected genes, technique of expression profiling and selected antibodies in case of IHC need to be defined [162].

Different approaches can be used to overcome/complement the limitations of molecular subtyping. Intra-tumor heterogeneity is not entirely depicted by tumor bulk profiling but can be studied by single-cell profiling, spatial imaging techniques or imaging mass cytometry, such as cytometry by time-of-flight (CyToF). Inter-tumor heterogeneity and plasticity during tumor development, instead, can be analyzed by sequential analysis and repeated biopsies during patients’ history. More molecular subtyping studies are needed to increase the number of patient sequencing datasets. This is crucial to identify more classes of BLCa, particularly rare subtypes, and investigate the evolution of heterogeneous tumors at higher resolution.

Before considering the use of molecular subtypes in a daily clinical practice, a better functional understanding is necessary. The behavior of molecular subtypes can be functionally studied in pre-clinical models from patient-derived material such as patient-derived organoids, ex vivo tissue cultures and xenografts. Patient-derived models that reflect the molecular phenotype of the different subtypes can be employed to investigate the biology of BLCa subtypes and evaluate differential drug responses in the context of different subtypes. Interestingly, there is the possibility to genetically engineer these models to study the relevance of specific mutations in a given subtype. Moreover, this technology can be used to better understand how the different subtypes evolve and progress through the two-pathway model.

Do we really need to go so far to treat each patient individually or is some sort of grouping necessary for clinical trials? There is an urgent need for clinical trials that investigate the impact of molecular classification.

## 7. Conclusions

To conclude, the management of BLCa is complex and current classification systems do not depict the heterogeneity of this disease. Promising findings from the molecular investigation of BLCa cohorts show that grouping patients into classes based on the molecular landscape can be used to understand the pathogenesis of this complex disease. Early data suggests that these investigations can also be used to predict the response to treatment and its implementation should be taken into consideration for the future management of this disease. Ideally, to have reliable results, we should complement mRNA with IHC analysis of protein for each patient’s tumor. However, to move towards precision medicine, the understanding of the biological meaning of these molecular classes, the differences in the composition of cell subpopulations and clinical trials that investigate the impact of molecular classification will be mandatory.

## Figures and Tables

**Figure 1 ijms-21-05670-f001:**
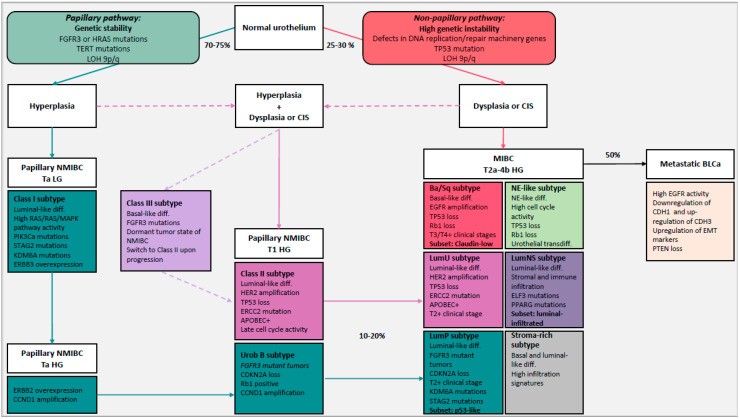
Simplified representation of the evolution of bladder cancer (BLCa) including the two distinct carcinogenic pathways (papillary and non-papillary pathways) and molecular subtypes of BLCa with their different characteristics. 75–80% of BLCa are non-muscle invasive bladder cancer (NMIBC) and 20–25% are muscle invasive bladder cancer (MIBC), of which 50% progress to metastatic BLCa. The papillary pathway includes hyperplasia which will give rise to Ta low-grade (LG) NMIBC and Ta/T1 high-grade (HG) NMIBC after the acquisition of mutations such as fibroblast growth factor receptor 3 (FGFR3) mutations or RAS mutations, loss of heterozygosity (LOH) 9p/q and telomerase reverse transcriptase (TERT) gene-promoter mutations. Class I/Urobasal A (Uro A) subtypes represent Ta LG NMIBC tumors and Uro B tumors were defined to be their progressed version. 10–20% of NMIBC potentially progress to MIBC and given the elevated expression of FGFR3 and the homozygous cyclin dependent kinase inhibitor 2A (CDKN2A) deletions in luminal papillary (LumP) tumors, they can originate from the progression of Class I/Uro A tumors through the papillary pathway. In the non-papillary pathway, instead, flat dysplasia and/or carcinoma in situ (CIS) are believed to be the precursors of MIBC tumors, after the gradual accumulation of genomic abnormalities such as loss of tumor suppressors TP53, defects in DNA replication/repair machinery genes (e.g., excision repair 2 (ERCC2)) and LOH 9q/p. Basal/Squamous (Ba/Sq) subtype represents the fraction of MIBC that originate directly from the non-papillary pathway. Given their genomic instability, some HG NMIBC tumors seem to derive from the co-occurrence of hyperplasia and dysplasia shown by the dashed arrows. Class II subtypes represent T1 HG NMIBC tumors that originate from the co-occurrence of hyperplasia and dysplasia, and luminal unstable (LumU)/genomic unstable (GU) tumors are the progressed version that advanced through the non-papillary pathway. Luminal non-specified (LumNS) exhibits features of the LumP and LumU subtypes. Class III represents a dormant tumor state of NMIBC that seems to switch to Class II upon progression. However, the origin of Class III tumors is not fully clear. It is probably a fraction of HG tumors that originated from both papillary and non-papillary pathways given the basal phenotype and FGFR3 mutations. The neuroendocrine-like (NE-like) subtype expresses neuroendocrine differentiation markers and its origin is not fully clear. They maybe originate from transdifferentiation of urothelial cells upon treatment. The stroma-rich subtype, associated with high infiltration signatures, such as LumNS, represents a heterogeneous class of tumors that we need to investigate at a higher resolution. Blue arrow indicates reoccurrence of NMIBC. Dashed arrows are hypothetical. Abbreviation: PI3KCa: phosphatidylinositol-4,5-bisphosphate 3-binase catalytic subunit alpha, STAG2: stromal antigen 2, KDM6A: lysine demethylase 6A, ERBB2/3: Erb-B2 receptor tyrosine kinase 2/3, CCND1: cyclin D1, EGFR: epidermal growth factor receptor, HER2: human epidermal growth factor receptor 2, APOBEC: apoliprotein B mRNA-editing enzyme catalytic polypeptide-like, PPARG: peroxisome proliferator activated receptor gamma, ELF3: E74 Like ETS transcription factor 3, CDH1: cadherin 1, CDH3: cadherin 3, EMT: epithelial-to-mesenchymal transition, PTEN: phosphatase and tensin homolog, diff.: differentiation, inf: infiltration.

**Figure 2 ijms-21-05670-f002:**
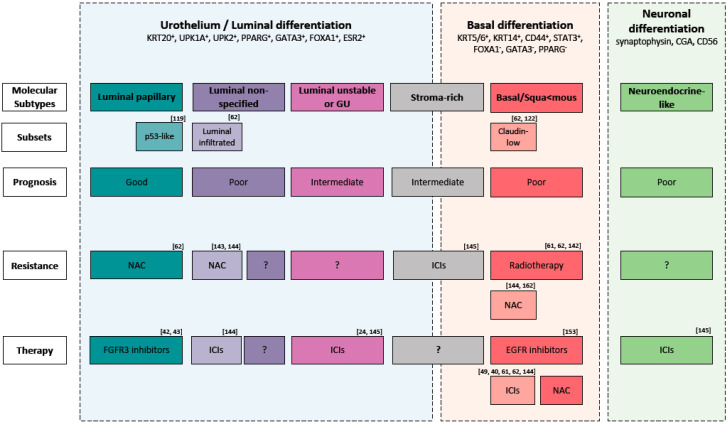
Molecular subtypes of muscle-invasive bladder cancer and their main expression markers, possible molecular subsets, prognosis, therapy resistance and therapy sensitive. Abbreviation: GU: Genomic Unstable, ICIs: immune checkpoint inhibitors, NAC: Neoadjuvant cisplatin-based chemotherapy.

**Table 1 ijms-21-05670-t001:** Molecular subtypes of non-muscle invasive bladder cancer and their main features.

NMIBC Subtypes	Differentiation	Oncogenic Mechanism	Molecular Features	Stage/Grade	Associated Risk
**Class I**	Luminal	FGFR3 mutations	ERBB3 overexpression High RAS/RAF/MAPK pathway activity Early cell cycle regulators (CCDN1 amplification)	Low stage and grade	High risk of reoccurrence
**Class II**	Luminal	HER2 amplification	TP53/RB1 loss ERCC2 mutations Late cell cycle regulators EMT transcript factors APOBEC mutation signatures	High stage and grade	High risk of progression
**Class III**	Basal	FGFR3 mutations	Dormant tumor state of NMIBC that switch to Class II upon progression	High stage and grade	High risk of progression

Abbreviation: NMIBC: non-muscle invasive bladder cancer, EMT: epithelial-to-mesenchymal transition.

**Table 2 ijms-21-05670-t002:** Molecular subtypes of muscle invasive bladder cancer and their main features.

MIBC Subtypes	Differentiation	Oncogenic Mechanism	Molecular Features	Stage/Grade	Subsets
**LumP**	Luminal	FGFR3 mutations CDKN2A deletions	High RAS/RAF/MAPK pathway activity KDM6A mutations STAG2 mutations	T2+	p53-like: WT p53 gene expression signature
**LumNS**	Luminal	PPARG mutations	ELF3 mutations Elevated stromal and moderate immune infiltration	T2+	Luminal-infiltrated
**LumU**	Luminal	HER2 amplification Genomic instability	TP53 loss ERCC2 mutations High cell cycle activity APOBEC+ PPARG mutations	T2+	
**Ba/Sq**	Basal and squamous	EGFR mutations	TP53 and/or RB1 loss STAT3 regulon activation Elevated HIF1A activity	T3/T4	Claudin-low: Decreased expression of claudin-3 and claudin-4 Enrichment for TIC Overexpression of EMT transcript factors Stroma and immune infiltration High risk of metastatic disease
**Stroma-rich**	Basal and luminal		Elevated stroma and immune cells infiltration		
**NE-like**	Neuroendocrine	TP53 and/or RB1 loss	High cell cycle activity Urothelial transdifferentiation		

Abbreviation: MIBC: non-muscle invasive bladder cancer, LumP: luminal papillary, LumNS: luminal non-specified, LumU: luminal unstable, Ba/Sq: basal squamous, NE-like: neuroendocrine, STAT3: signal transducer and activator of transcription 3, HIF1A: hypoxia inducible factor 1 subunit alpha, TIC: tumor-initiating cells.

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
