# Peer review of "Evolution of Urothelial Bladder Cancer in the Context of Molecular Classifications"

_ijms, 2020, doi:10.3390/ijms21165670_

Round 1

Reviewer 1 Report

The article deals with a very interesting topic, but in a somewhat confusing way. The authors say they want to split the article into two parts, but that doesn't make sense based on the topic of the article.

1) It would be better to significantly reduce (if not eliminate) paragraphs 2.1 and 2.2. If the authors decide to keep them, it is necessary to reduce them by balancing the extent of the discussion on BCG (too long) compared to the therapy for metastatic disease

2) Paragraph 4 is the most important part of the article. However, it should be improved, perhaps adding a table in which the different classifications are compared. Currently the differences are only hinted at in Figure 1, but in an unclear way

3) Page 1 line 33: "tabacco" is an Italian word; please modify

4) Figure 2: "Squa<mous"; please modify

Author Response

Answer to Reviewer1’s comments:

1. Comment from reviewer 1: The article deals with a very interesting topic, but in a somewhat confusing way. The authors say they want to split the article into two parts, but that doesn't make sense based on the topic of the article.

Author response: We thank the reviewer for point this out. To address this, we modified the introduction of the review as follow:

“In this review we will present the state of the art, clinical guidelines and standard operating procedures of BLCa and elucidate their limitations. We will also discuss how molecular subtypes discovered from the profiling of tumors samples improved our understanding of BLCa and their potential clinical meaning and application.” (line 71-74).

2. Comment from reviewer 1: It would be better to significantly reduce (if not eliminate) paragraphs 2.1 and 2.2. If the authors decide to keep them, it is necessary to reduce them by balancing the extent of the discussion on BCG (too long) compared to the therapy for metastatic disease.

Author response: We thank the reviewer for the comment, but we think that paragraphs 2.1 and 2.1 are important to introduce the standard operation procedures of bladder cancer and their limitations therefore we reduced them to balance the extent of the discussion. (line 77-149, page 2-3)

3. Comment from reviewer 1: Paragraph 4 is the most important part of the article. However, it should be improved, perhaps adding a table in which the different classifications are compared. Currently the differences are only hinted at in Figure 1, but in an unclear way.

Author response: We thank the reviewer for the suggestion, and we included two tables with the most important features of the molecular subtypes, table 1. for non-muscle invasive subtypes (page 8) and table 2.  (page 9) for muscle invasive subtypes.

4. Comment from reviewer 1: Page 1 line 33: "tabacco" is an Italian word; please modify.

Author response: We changed “tabacco” with “tobacco” (line 33).

5. Comment from reviewer 1: Figure 2: "Squa<mous"; please modify

Author response: We corrected “Squamous” in Figure 2 (page 12). 

Reviewer 2 Report

Minoli and Colleagues describe in this nice, interesting and comprehensive review the molecular background, evolution of Urothelial bladder cancer (UBC) and molecular subclassification. The manuscript is well written, easy to read and gives a very interesting overview.

The manuscript starts with the clinical problem and therapeutic background of UBC coming to the problem of missing predictive biomarkers (part 1-3). In the next part (part 4) the molecular background and different molecular subtypes with respective markers are introduced.  This is followed by a part describing possible clinical implications of molecular subtypes (part 5). At the end of the manuscript (part 6) problems with molecular subtypes and potential ways of further development are discussed.

In my opinion this is an interesting study, deserves to be published. However I have some comments to further improve the work.

  • I would suggest to rename part six (How to Overcome the Limitations of Molecular Classification) in „Limitations and perspectives of molecular classification” and introduce or move some additional aspects in this chapter.

In my opinion the following aspects are important with regards to limitations and further perspectives of molecular subtypes.

  1. First the preanalytical and analytical aspects of molecular classification (Tumor cell content; Immune cell infiltration, selected genes, technique of expression profiling, Selected antibodies in case of IHC) [Sjödahl 2019 J. Pathol]. In summary the lack of standardization which is clearly needed for implementation in the clinical diagnostic setting.
  2. From my understanding it is important to mentioned that molecular subtypes are established mainly in untreated tumor samples and that these fact is important for prognostic data interpretation.
  3. With regards to the prognostic value of the molecular subtypes some discrepant result or not so significant papers should be mentioned and introduced (e.g. Kollberg 2019 Urologic Oncology). Suggested that the prognostic value of a specific subtype clearly needs further validation.

Author Response

Answer to Reviewer2’s comments:

1. Comment from reviewer 2: Minoli and Colleagues describe in this nice, interesting and comprehensive review the molecular background, evolution of Urothelial bladder cancer (UBC) and molecular subclassification. The manuscript is well written, easy to read and gives a very interesting overview.

The manuscript starts with the clinical problem and therapeutic background of UBC coming to the problem of missing predictive biomarkers (part 1-3). In the next part (part 4) the molecular background and different molecular subtypes with respective markers are introduced.  This is followed by a part describing possible clinical implications of molecular subtypes (part 5). At the end of the manuscript (part 6) problems with molecular subtypes and potential ways of further development are discussed.

In my opinion this is an interesting study, deserves to be published. However I have some comments to further improve the work.

Author response: We thank the reviewer

2. Comment from reviewer 2: I would suggest to rename part six (How to Overcome the Limitations of Molecular Classification) in “Limitations and perspectives of molecular classification”

 Author response: As suggested by the reviewer, we changed the title of chapter 6 in “Limitations and Perspectives of Molecular Classification.”

3. Comment from reviewer 2: In my opinion the following aspect are important with regards to limitation and further perspective of molecular subtypes.

First the preanalytical and analytical aspects of molecular classification (Tumor cell content, Immune cell infiltration, selected genes, technique of expression profiling, Selected antibodies in case of IHC) [Sjödahl 2019 J. Pathol]. In summary the lack of standardization which is clearly needed for implementation in the clinical diagnostic setting.

Author response: We agree with the reviewer’s assessment. Accordingly, to point out the lack of standardization we revised the text as follows:

 “There is current standardization and consensus on gene expression based molecular subtyping [96]. However, further standardizations are needed for the implementation in the clinical setting. In addition other analytical aspects such as tumor cell content, immune cell infiltration, selected genes, technique of expression profiling, and selected antibodies in case of IHC need to be defined [162]. ”   (line 619-623)

4. Comment from reviewer 2: From my understanding it is important to mention that molecular subtypes are established mainly in untreated tumor samples and that these facts is important for prognostic data interpretation.

Author response: We agree with the reviewer’s assessment and we revised the text as follows:

“Of utmost importance, these molecular classes have been established based on investigations of mainly untreated tumor samples.” (line 612-613).

5. Comment from reviewer 2: With regards to the prognostic value of the molecular subtypes some discrepant result or not so significant papers should be mentioned and introduced (e.g. Kollberg 2019 Urologic Oncology). Suggested that the prognostic value of a specific subtype clearly needs further validation.

Author response: We thank the reviewer for the suggestion. To address this problem, we explained that nowadays there is a discrepancy in outcome of patients with luminal-like tumors and that further validation is necessary. We added the following statement:

“Discrepancy in outcome of patients with luminal-like tumors has been reported [92, 121, 132]. ”

Reviewer 3 Report

The authors reviewed the problems of current management and precision medicine the understanding of the biological meaning of these molecular classes, the differences in the composition of cell subpopulations of BLCa.

Well written.

This is an interesting and impressed report. The contents are concise and easy to read.

The period at line 420 on page 9 is missing.

Author Response

Answer to Reviewer3’s comments:

1. Comment from the reviewer 3: The authors reviewed the problems of current management and precision medicine the understanding of the biological meaning of these molecular classes, the differences in the composition of cell subpopulations of BLCa.

Well written.

This is an interesting and impressed

Author response: We thank the reviewer.

2.Comment from the reviewer 3: The period at line 420 on page 9 is missing.

Author response: We added the period at the line 452.

Round 2

Reviewer 1 Report

The article has been significantly modified. Now the manuscript is more fluent and the topic is treated in a more orderly way.
I report only a small correction to do before publishing it: on page 1, the authors decided to use the acronym BLCa to indicate bladder cancer.

Why do they specify it again in pages 2 line 30 (paragraph 2 - title) ?

And why do they no longer use the acronym on page 3 line 10 (paragraph 2.2 line 3 "advanced bladder cancer")?

After making these small changes the article can be published

Author Response

Answer to Reviewer1’s comments:

1. Comment from reviewer 1: The article has been significantly modified. Now the manuscript is more fluent and the topic is treated in a more orderly way.

 Author response: We thank the reviewer.

2. Comment from reviewer 1: I report only a small correction to do before publishing it: on page 1, the authors decided to use the acronym BLCa to indicate bladder cancer.

Why do they specify it again in pages 2 line 30 (paragraph 2 - title) ?

And why do they no longer use the acronym on page 3 line 10 (paragraph 2.2 line 3 "advanced bladder cancer")?

After making these small changes the article can be published

Author response: We thank the reviewer for point this out. We exchanged  “bladder cancer” with the acronym BLCa in the title of chapter 2 (page 2 line 74) and at page 3 line 107.